# The Importance of M1-and M2-Polarized Macrophages in Glioma and as Potential Treatment Targets

**DOI:** 10.3390/brainsci13091269

**Published:** 2023-08-31

**Authors:** Jiangbin Ren, Bangjie Xu, Jianghao Ren, Zhichao Liu, Lingyu Cai, Xiaotian Zhang, Weijie Wang, Shaoxun Li, Luhao Jin, Lianshu Ding

**Affiliations:** 1Department of neurosurgery, The Affiliated Huaian No. 1 People’s Hospital of Nanjing Medical University, Nanjing Medical University, Huai’an 223000, China; renjb1847493134@163.com (J.R.); xubangjie0919@163.com (B.X.); zhichaoliu77@163.com (Z.L.); cailingyu2021@163.com (L.C.); 15195307883@163.com (X.Z.); weijie_wang83@163.com (W.W.); somebodylsx@163.com (S.L.); 18052995606@163.com (L.J.); 2Department of Thoracic Surgery, Shanghai Chest Hospital, Shanghai Jiaotong University, Shanghai 200030, China; jhren2020@163.com

**Keywords:** glioma, glioblastoma, tumor-associated macrophage, polarization, treatment

## Abstract

Glioma is the most common and malignant tumor of the central nervous system. Glioblastoma (GBM) is the most aggressive glioma, with a poor prognosis and no effective treatment because of its high invasiveness, metabolic rate, and heterogeneity. The tumor microenvironment (TME) contains many tumor-associated macrophages (TAMs), which play a critical role in tumor proliferation, invasion, metastasis, and angiogenesis and indirectly promote an immunosuppressive microenvironment. TAM is divided into tumor-suppressive M1-like (classic activation of macrophages) and tumor-supportive M2-like (alternatively activated macrophages) polarized cells. TAMs exhibit an M1-like phenotype in the initial stages of tumor progression, and along with the promotion of lysing tumors and the functions of T cells and NK cells, tumor growth is suppressed, and they rapidly transform into M2-like polarized macrophages, which promote tumor progression. In this review, we discuss the mechanism by which M1- and M2-polarized macrophages promote or inhibit the growth of glioblastoma and indicate the future directions for treatment.

## 1. Introduction

Glioma is the most common and malignant tumor of the central nervous system, accounting for approximately 46% of all intracranial tumors [1]. Among these tumors, glioblastoma (GBM)—classified as a WHO grade IV glioma—is the most aggressive, with poor prognosis and no effective treatment currently. GBM derived from the neuroepithelium is located in the subcortical area with invasive growth, and the incidence of GBM increases with age. Because of its high invasiveness, metabolic rate, and heterogeneity, patients usually survive for no more than 15 months with a 5-year survival rate of less than 6.8% despite aggressive surgical resection and chemoradiotherapy [2,3]. 

The tumor microenvironment (TME) containing the extracellular matrix (ECM), parenchyma cells, soluble factors, infiltrating immune cells, glial cells, glioma stem cells, vascular cells, fibrotic cells, and adipocytes plays a critical role in tumor growth and immune evasion [4,5]. Recent studies have provided substantial evidence that the progression of GBM depends on various cell-to-cell interactions and inflammatory factors in the TME, and that the components of the TME are regulated by signaling pathways and secreted factors of GBM [6,7]. The TME contains a lot of tumor-associated macrophages (TAMs), approximately 33%, which correlate with both the GBM phenotype and tumor grade [1,8]. Some have pointed out that TAM has two origins: (a) from bone marrow-derived macrophages (BMDMs) named glioblastoma-associated macrophages and (b) from resident macrophages/microglia named glioblastoma-associated microglia [1]. Many studies have shown that TAM can promote tumor proliferation, invasion, metastasis, and angiogenesis and indirectly promote an immunosuppressive microenvironment. 

Functionally, TAMs are divided into tumor-suppressive M1-like (classic activation of macrophages) and tumor-supportive M2-like (alternatively activated macrophages) polarized cells, with the latter occupying the majority in GBM [9]. M1 TAMs are usually activated by interferon gamma (IFN-γ), lipopolysaccharides (LPS), tumor necrosis factor (TNF-α), NADPH-oxidase, inducible nitric oxide synthase (iNOS), and immune receptors TLR2/4, which promote T helper 1 (Th1) responses, and also activate natural killer (NK) cells through pro-inflammatory cytokines [1,10,11,12,13]. These cytokines include TNF-α, IL-1β, IL-6, IL-8, IL-12, and IL-23, all of which inhibit tumor growth [1,10]. M2 TAMs are activated by peroxisome proliferator-activated receptor-γ (PPARγ) and STAT6, an inhibitor of the NF-κB pathway [14]. M2 TAMs secrete anti-inflammatory cytokines, including arginase 1 (ARG1), interleukin-13 (IL-13), IL-10, IL-4, vascular endothelial growth factor (VEGF), and TGF-β to activate a Th2-type immune response and promote tumor progression by stimulating angiogenesis, maintaining tumor cell stemness, facilitating immune infiltration, remodeling tissues, and inducing drug resistance [1,10,15,16,17] (Figure 1). Based on relevant articles, we have tabulated the markers of M1 and M2 polarization (Table 1).

Furthermore, M2-polarized macrophages can be further subdivided into M2a, M2b, M2c, and M2d, which are triggered and developed in different microenvironments [18,19]. M2a can stimulate the Th2 response, kill pathogens, induce allergy development, and are activated by interleukin IL-4, IL-13, or fungal and helminth infections [8,13,15]. Similar to M2a, M2b polarized by IL-1 receptor ligands or LPS plus immune complexes is involved in the Th2 response and immune regulation [8,13,20]. The M2c population—stimulated by IL10, TGF-β, and glucocorticoids—promotes tumor growth and is involved in immunoregulation, matrix deposition and tissue remodeling [1,12,15]. M2d, which acts as a switching macrophage or angiogenic “M2-like” phenotype and is polarized by IL-6 and adenosines, determines the expression levels of the M2d markers VEGF-A, IL-10, IL-12, TNF-α, and transforming growth factor-β (TGF-β) [8,21,22,23]. 

This review aims to explore the mechanism by which M1- and M2-polarized macrophages promote or inhibit the growth of glioblastoma and discuss the corresponding treatments.

**Table 1 brainsci-13-01269-t001:** The markers of M1 and M2 TAMs.

M1 TAMs Markers			M2 TAMs Markers		
Study First Author	Year of Publication	Marker	Study First Author	Year of Publication	Marker
Kang et al. [11]	2017	iNOS	Codrici et al. [1]	2022	MRC1
Peng et al. [24]	2022		Yang et al. [9]	2018	
Xiao et al. [25]	2022	TSPO	Zhang et al. [26]	2022	
Codrici et al. [1]	2022	IL-12B	Codrici et al. [1]	2022	CD163
Zhang et al. [26]	2022		Kang et al. [11]	2017	
Zhang et al. [26]	2022	IL-12A	Bianconi et al. [13]	2022	
Kang et al. [11]	2017	CD40	Peng et al. [24]	2022	
Bianconi et al. [13]	2022		Jiang et al. [27]	2022	
Hambardzumyan et al. [15]	2016		Zhang et al. [28]	2022	
Xiao et al. [25]	2022	IL-1β	Hansen et al. [29]	2022	Timp1
Jiang et al. [27]	2022		Umemura et al. [30]	2008	CD36
Umemura et al. [30]	2008		Xia et al. [31]	2020	1IDO1
Xiao et al. [25]	2022	IL-6	Umemura et al. [30]	2008	CCL17
Curtale et al. [32]	2019		Curtale et al. [32]	2019	
Liu et al. [33]	2022	CD80	Sanders et al. [34]	2021	CCL4
Bianconi et al. [13]	2022	CD74	Peng et al. [24]	2022	VEGFA
Peng et al. [24]	2022	CD86	Hansen et al. [29]	2022	
Xiao et al. [25]	2022		Xia et al. [31]	2020	EGF
Curtale et al. [32]	2019		Codrici et al. [1]	2022	Fizz1
Kang et al. [11]	2017	CD197	Yang et al. [9]	2018	
Zhang et al. [26]	2022	IL-23A	Bianconi et al. [13]	2022	
Liu et al. [33]	2022		Peng et al. [24]	2022	
Codrici et al. [1]	2022	NOS2	Codrici et al. [1]	2022	Arg-1
Zhang et al. [26]	2022		Yang et al. [9]	2018	
Zhang et al. [28]	2022		Kang et al. [11]	2017	
Bianconi et al. [13]	2022	MHC-II	Bianconi et al. [13]	2022	
Xiao et al. [25]	2022	CXCL9	Peng et al. [24]	2022	
Sanders et al. [34]	2021		Zhang et al. [26]	2022	
Xiao et al. [25]	2022	CXCL10	Jiang et al. [27]	2022	
Zhang et al. [26]	2022		Hansen et al. [29]	2022	
Umemura et al. [30]	2008		Peng et al. [24]	2022	IL-10
Jiang et al. [27]	2022	TNF-α	Zhang et al. [26]	2022	
Umemura et al. [30]	2008		Hansen et al. [29]	2022	
Zhang et al. [28]	2022	IRF-5	Zhang et al. [26]	2022	RETNLB
Xiao et al. [25]	2022	CCL5	Jiang et al. [27]	2022	TGF-β
Sanders et al. [34]	2021		Liu et al. [33]	2022	
Codrici et al. [1]	2022	Ciita	Zhang et al. [28]	2022	MS4A4A
Sanders et al. [34]	2021	p-STAT1	Zhang et al. [28]	2022	VSIG4
			Bianconi et al. [13]	2022	p-STAT3
			Yang et al. [9]	2018	Mgl-2
			Biswas et al. [35]	2008	
			Codrici et al. [1]	2022	Chi3l3
			Codrici et al. [1]	2022	SOCS2
			Xia et al. [31]	2020	Ym1
			Biswas et al. [35]	2008	
			Liu et al. [33]	2022	MSR1
			Yang et al. [9]	2018	CD11c
			Codrici et al. [1]	2022	CD204
			Bianconi et al. [13]	2022	
			Sanders et al. [34]	2021	
			Kang et al. [11]	2017	CD206
			Bianconi et al. [13]	2022	
			Peng et al. [24]	2022	
			Jiang et al. [27]	2022	
			Umemura et al. [30]	2008	
			Peng et al. [24]	2022	CD68
			Codrici et al. [1]	2022	CCL2
			Xiao et al. [25]	2022	
			Peng et al. [24]	2022	CCL18
			Xiao et al. [25]	2022	
			Curtale et al. [32]	2019	
			Xia et al. [31]	2020	CCL20
			Umemura et al. [30]	2008	CCL22
			Xia et al. [31]	2020	

## 2. The Mechanism by Which M1- and M2-Polarized Macrophages Promote or Inhibit the Growth of Glioblastoma

### 2.1. M1 and M2 Polarization in Different Stages and Parts of Tumors

M1 and M2 polarization varies at different stages of tumor progression. TAMs exhibit an M1-like phenotype in the initial stages of tumor progression, and along with the promotion of tumor lysis and the functions of T cells and NK cells, tumor growth is suppressed [18,35,36]. In the initial stages of GBM formation, MDMs characterized by chemokine receptor 2 (CCR2) are recruited to perivascular areas while CX3CR1-enriched TAMs are recruited to the peritumoral regions [37]. However, when tumor environment accounts for a switch in macrophage phenotype favoring a pro-invasive and immunosuppressive M2 state until a critical point, TAMs rapidly transform into M2-like cells, promoting tumor progression [35,36,38]. Admittedly, there are not enough studies to prove this conclusion; therefore, it is difficult to determine the stage of the M2 phenotype in tumor progression.

Due to immune cell heterogeneity between the core and peripheral regions of GBM, the distribution of M1 and M2 TAMs is also different [18,39]. Generally, the core regions of GBM are more hypoxic and acidic than the peripheral regions, which increases the aggressiveness of GBM [40]. HIFs (hypoxia inducible factor-1) activated by tumor-induced hypoxia through various mechanisms lead to the polarization of TAMs, where M2 TAMs are usually enriched in the core hypoxic zone, whereas M1 is enriched in the peripheral normoxic zones [18,33,41].

### 2.2. Galectin and AKT/GSK3β/IRF1/Gal-9/Tim3 Pathway

Lectins expressed by immune cells function to recognize physiological and tumor-associated carbohydrates and bind to specific glycosylation structures [42,43]. Lectins, which can be incorporated into the cellular membrane or have a soluble form, can be divided into C-type, I-type, P-type, S-type lectins (also known as galectins), and pentraxins, based on their subcellular location and structures [43,44]. Structurally, galectins can be differentiated in dimeric form (including Gal-1, 2, 5, 7, 10, 11, 13, 14, 15), tandem-repeat form composed of at least two carbohydrate CRDs (including Gal-4, 6, 8, 9), and the monomer or multivalent chimera type (Gal-3) [43,45]. Currently, because of their capacity to modulate adaptive immune responses, Gal-1, Gal-3, and Gal-9—known as biomarkers of poor cancer prognosis—have been widely studied [46,47]. *LGALS1*, the gene encoding Gal-1, promotes M2 TAMs secretion of IL-10 and TGFβ, which contribute to a tumor-supporting and immunosuppressive environment [48]. Another study found that the downregulation of Gal-1 reduces TAMs switching from M1 to M2 in GBM [49]. Recently, Woensel et al. designed siRNA targeting Gal-1 (siGal-1)-loaded chitosan nanoparticles to help nose-to-brain transport, then silencing Gal-1 in the TME, which reduces the macrophage polarization switch from M1 (pro-inflammatory) to M2 (anti-inflammatory) [49].

Gal-9, known as an eosinophil chemoattractant and negative modulator of the adaptive immune response, is inhibited by α-lactose, positively related to M2 TAMs, and binds with T cell immunoglobulin and mucin domain 3 (Tim3) to inhibit the immune microenvironment in GBM [43,50,51]. Tim3 is expressed in NK cells, monocytes, macrophages, dendritic cells, mast cells, and Th1 and Th17 cells and binds with Gal-9 to inhibit the polarization of Th17 [46,52]. Zhu et al. found that the mechanism of interaction between Tim-3 and Gal-9 is the inhibition of TH1 immunity by the selective deletion of Tim-3^+^ TH1 cells [53]. 

Phosphatase and tensin homolog (PTEN)—a widely expressed tumor suppressor—frequently shows genomic deletions in brain, bladder, and prostate tumors [54,55]. PTEN negatively regulates the phosphatidylinositol 3 kinase (PI3K)–AKT-rapamycin (mTOR) signaling pathway and is involved in cell proliferation, apoptosis, survival, and metabolism [54,56]. Furthermore, PTEN-deficient GBM cells activate the PI3K/AKT pathway, which suppresses interferon regulatory factor 1 (IRF1) degradation by phosphorylating glycogen synthase kinase 3 beta (GSK3β) into its inactive form, with higher levels of IRF1, inducing Gal-9 transcription (Figure 2). Additionally, the Gal-9/Tim3 interaction promotes the polarization of M2 TAMs, which promotes glioma angiogenesis through VEGFA secretion [57].

### 2.3. Chemerin/CMKLR1 Axis

Chemerin, also known as retinoic acid receptor responder 2 (RARRES2) or tazarotene-induced gene 2 (TIG2), is a secreted protein of 163 amino acids, and its active forms are produced via C-terminal processing [58,59,60]. It has been shown that chemerin participates in tissue inflammation, glucose homeostasis, atherosclerosis of the arteries, diabetic kidney disease (DKD), the progression of various nausea tumors, mediating the formation of blood vessels, and stimulating vascular smooth muscle cell (VSMC) proliferation as well as carotid intimal hyperplasia [61,62]. Additionally, chemerin is highly expressed in white adipose tissue and the liver and lungs [63,64]. The chemerin can be combined with three receptors: G protein-coupled receptor chemokine-like receptor 1 (CMKLR1 or ChemR23), G protein-coupled receptor-1 (GPR1), and chemokine receptor-like 2 (CCRL2). The first two regulate the biological activities of chemerin isoforms [65,66]. CMKLR1 transcription is reportedly enhanced by LPS and IFN-γ, which induce the polarization of monocytes to M2 macrophages [58,67]. Chemerin has two distinct functions in cancer, including either promoting or inhibiting tumors, depending on different mechanisms [58].

In atherosclerosis, NF-κB activated by chemerin via the MAPK and PI3K/Akt pathways promotes transcriptional activation of gene promoter regions, and highly expressed adhesion molecules in endothelial cells foster the initiation of atherosclerosis. Moreover, the augmentation of atherosclerotic plaque formation and progression is associated with M2 macrophages [62,68]. In diabetic kidney disease (DKD), Wang et al. found that chemerin can enhance the TGF-β1/SMAD/CTGF signaling pathway both in vitro and in vivo, thereby promoting the development and progression of DKD and causing a significant reduction in renal function [69]. It has also been reported that chemerin activates the p38 MAPK pathway, which promotes inflammation and kidney injury [70]. In angiogenesis, chemerin binding to CMKLR1 leads to endothelial cell apoptosis and vessel regression, which is mediated through the activation of PTEN, inhibition of the PI3K/AKT pathway, and enhanced FOXO1 activity (PTEN/AKT/FOXO1 axis) [60]. Furthermore, chemerin in the bone marrow promotes osteogenic differentiation and bone formation through the AKT/GSK3β/β-catenin axis, which provides a reference scheme for the treatment of osteoporosis [63].

Chemerin is also expressed in malignant tumors such as GBM, and a study by Wu et al. demonstrated that patients with GBM have high levels of chemerin expression in both the tumor and serum, which was inversely associated with patient survival [61]. Furthermore, chemerin significantly enhanced the migration and invasion abilities of GSCs, suggesting an enhancing effect on the mesenchymal features of GBM cells. Chemerin is positively correlated with TNF-α expression, which is indispensable for the pro-mesenchymal effect of chemerin in GBM cells. Chemerin binding to CMKLR1 activates the NF-κB pathway, which promotes the infiltration of TAMs and M2 polarization (Figure 2). The M2 TAMs induced by chemerin enhanced the pro-mesenchymal capacity of TAMs [61,71]. Additionally, another study suggested that M2 tumor-associated macrophages promote tumor growth via the NF-κB/IL-6/STAT3 signaling pathway [13].

Therefore, targeting the chemerin/CMKLR1 axis in GBM to inhibit NF-κB is expected to be a new treatment for suppressing the progression of GBM.

### 2.4. MTA/A_2B_ Receptor/M2 Pathway

The homozygous deletion of cyclin-dependent kinase inhibitor 2A/B (CDKN2A/B) at the 9p21 chromosome occurs in the early stages of GBM, and since methylthioadenosine phosphorylase (MTAP) is close to the CDKN2A tumor suppressor sites, this leads to the co-deletion of MTAP [72,73]. MTAP is involved in the salvage pathways of both methionine and adenine, catalyzing the conversion of methylthioadenosine (MTA); therefore, the deletion of MTAP often leads to the accumulation of MTA [29,74]. Reportedly, MTA can suppress immunity by downregulating TNF-α through binding to adenosine receptors, and it is widely used in the treatment of colitis, hepatitis, and encephalitis [29,75]. A_2B_ adenosine receptor (AR), activated by adenosine and highly expressed in GBM, regulates GBM cell apoptosis, proliferation, and immunity [76]. Additionally, a study by Kitabatake et al. found that adenosine induced by CD73 activated the A_2B_ receptor, which promoted recovery from DNA damage, cell migration, and actin remodeling [77].

Hansen et al. found that MTAP-deficient GBM patients overexpressed MTA binding to A_2B_ receptor, which functions downstream of ARs like STAT3, inducing upregulation of ARG-1 and IL-10, the markers of M2 TAMs (Figure 3). Furthermore, MTA activates the expression of VEGFA, suggesting that MTA activated the M2d subtype. The polarization of M2 TAMs is regulated by the transcription factor C/EBP (CEBPA/CEBPB) [29].

Furthermore, in addition to central nervous system tumors, Ludwig et al. reported that tumor-derived exosomes (TEX) produced by HPV^+^ head and neck squamous cells induce A_2B_ receptor signaling, resulting in the polarization of M2-like macrophages [78]. Ito et al. found that MTA induces the accumulation of M2 TAMs participating in wound healing and tissue repair processes [79]. In the study by Takei et al., M2 macrophages were also found to be involved in the initial phases of MTA-capped pulp tissue healing [80].

### 2.5. HMGB1/RAGE/NF-κB/NLRP3/M1 Pathway

The high mobility group box 1 protein (HMGB1), a highly conserved chromosomal protein that binds to DNA and stabilizes nucleosomes, mainly located in the nucleus, cytoplasm, and extracellular [81,82,83]. HMGB1 is involved in the DNA damage response, gene transcription, autophagy, cell proliferation, inflammation, and autoimmunity [83,84]. Recently, studies on the role of HMGB1 in gliomas, especially glioblastomas, have been published in several journals. Endogenetic noncoding RNA, such as miR-339-5p, inhibit angiogenic mimicry, migration, and invasion of glioma cells by inhibiting the PTP4A1/HMGB1 signaling pathway [85]. In astrocytes, HMGB1 binding to the CpG islands of the SH3 domain-containing 1 (SASH1) gene participates in the methylation of the SASH1 gene, downregulating the expression of the SASH1 gene and integrin β8, thereby reducing cell adhesion and promoting cell migration [86]. In another study, P62 induced by HMGB1 augmented the degradation of GSK-3β to activate the GSK-3β/Snail pathway, thereby promoting Snail-mediated epithelial–mesenchymal transition (EMT) in GBM [87]. In addition, HGBM1 activates the receptors on GBMs via the NF-kB, IFN regulatory factor-3 (IRF3), and phosphoinositide 3-kinase (PI3K) pathways to activate tumor-associated dendritic cells (TADC), CD8^+^ T cells, and macrophages [88,89].

HMGB1 in GBM cells is secreted to the extracellular matrix via autophagic vacuoles, and then binds to RAGE, which activates the pathway of regulating cell differentiation, growth, motility, and death and especially in promoting the proliferation and invasion of tumor cells [81,82]. It has been reported that neutrophil extracellular traps (NETs) produced by tumor-infiltrating neutrophils (TINs) mediate the interaction of glioma and TAMs by regulating the HMGB1/RAGE/IL-8 axis [90]. It has also been found that hyperglycemia may participate in glioma growth and suppress anti-tumor immune responses by activating the HMGB1–RAGE axis [91]. 

Li et al. found that HMGB1 is transported from the nuclei to the cytoplasm and extracellular space via autophagic vacuoles, following treatment with TMZ, and GBM patients with high levels of HMGB1 in the intracellular region always have a worse prognosis. It was also found that HMGB1 activated RAGE through the phosphorylation of ERK1/2 and IKB, thereby activating the RAGE/NF-κB/NLRP3 inflammasome pathway—which promoted the release of TNF-α, IFN-γ, IL-1β, IL-6, IL-8, and CCL2—thereby enhancing the M1-like polarization of TAMs [81] (Figure 3). Thus, the HGBM1/RAGE axis may be an important target for glioma treatment in the future. Furthermore, according to recent studies, in microglial cells stimulated by S100 calcium-binding protein B (S100B), receptor for advanced glycation end products (RAGE) initiates the STAT3 pathway and inhibits the polarization of M1 TAMs, thus inhibiting the secretion of IL-1β and TNF-α [92].

### 2.6. BACE1/IL-6R/SIL6R/IL-6/STAT3/M2 Pathway

Aspartyl protease β-site AβPP-cleaving enzyme 1 (BACE1), which belongs to the family of proteases named β-secretases, is a transmembrane glycoprotein that acts as an aspartyl protease [93,94]. BACE1 is mainly expressed in the central nervous system and located on the cell surface and the membrane of intracellular vesicles [95]. BACE1 is an important target in the treatment of Alzheimer’s disease (AD), and it catalyzes the rate-limiting step in amyloid-β (Aβ) production [96,97], which exacerbates neuroinflammation, and promotes vascular and parenchymal damage. A study by Wang et al. demonstrated that leptin reduced the acetylation of the p65 subunit in a SIRT1-dependent manner, thereby decreasing the transcriptional activity of NF-κB, and downregulating BACE1, which is mediated by NF-κB and reduces amyloid-β genesis [97]. It has also been reported that BACE1 is involved in the proliferation, migration, and invasion of osteosarcoma via the miR-762/SOX7 axis and in the invasion and metastasis of hepatocellular carcinoma via the miR-377-3p/CELF1 axis [98,99].

In GBM cells, BACE1 acts as a transmembrane protease that mediates the shedding of IL-6R, and soluble IL-6 receptor (sIL-6R) in the extracellular matrix binds with IL-6, forming an IL-6/sIL-6R complex. It is well known that sIL-6R prolongs the half-life of IL-6 and stabilizes IL-6 signaling [100]. Additionally, the IL-6/sIL-6R complex binds to glycoprotein 130 (gp130), thus activating the phosphorylation of STAT3. Activated STAT3 signaling promotes the polarization of M2 TAMs to promote tumor progression, invasion, and migration (Figure 3). Zhai et al. also found that GBM patients with high levels of BACE1 always have worse prognosis and shorter survival, and the inhibition of BACE1 promotes a switch from M2 to M1 TAMs, which suppresses GBM growth. Furthermore, inhibition of BACE1 in combination with low-dose radiation has a more significant effect on the treatment of GBM [93].

### 2.7. The B2M/PIP5K1A/PI3K-AKT-MYC/TGF-β1/SMAD/M2 Pathway

Human MHC-I molecules, including HLA-A, -B, and -C, and non-classical HLA-E, -F, and -G, each of which consists of a specific MHC-encoded polymorphic heavy chain and β2-microglobulin (B2M), and B2M functions to load antigen peptides onto MHC-I molecules properly and maintain the cell surface localization of MHC-I [101,102,103]. B2M, a non-glycosylated protein, is mainly distributed in the cell membrane and cytoplasm of glioma cells and is highly expressed in GBM. Reportedly, highly expressed B2M can predict the poor prognosis of glioma patients and mediate immune cell infiltration via chemokines [104].

PIP5K1A, a phosphatidylinositol phosphate kinase located in the cell membrane and cytoplasm, can produce phosphatidylinositol 3,4,5-triphosphate (PIP3), which recruits and activates the serine/threonine protein kinase AKT, thereby specifically activating the PI3K/AKT pathway [105,106,107]. Some studies have confirmed that, in addition to GBM, PIP5K1A is also involved in human hepatocellular carcinoma (HCC) and non-small-cell lung cancer [108,109]. Li et al. showed that B2M colocalized with PIP5K1A in the membrane using confocal imaging in GBM cells, suggesting that B2M interacts with PIP5K1A in GSC. Furthermore, upon B2M knockdown, the localization of PIP5K1A shifted from the membrane to the cytoplasm [106]. MYC is downstream of the PI3K/AKT signaling pathway and is indispensable in the maintenance of cancer stem cells orchestrating their proliferation, apoptosis, differentiation, angiogenesis, immune evasion, and metabolism [110,111,112]. Since the active site of MYC is located in the promoter region of TGF-β1, MYC activates the expression of TGF-β1, promoting the polarization of M2 TAMs via SMAD and PI3K/AKT signaling. Li et al. also found that B2M expression was positively correlated with glioma grade and that high levels of B2M are related to the poor prognosis of GBM patients [106].

### 2.8. GC/GR/FGF20/FGFR1/β-Catenin Pathway

Fibroblast growth factors (FGFs), a paracrine cytokine that binds to heparan sulfate proteoglycan and fibroblast growth factor receptors (FGFRs), promotes cell proliferation and is involved in embryogenesis, tissue regeneration, and the healing of gastric mucosal damage associated with Helicobacter pylori infection [113,114]. FGF20, a member of the FGF family, is a direct target for β-catenin/TCF transcriptional regulation via LEF/TCF-binding sites, reportedly playing an important role in colorectal cancer and ovarian endometrioid adenocarcinoma [114,115]. FGFR family members—including FGFR1, FGFR2, FGFR3, FGFR4, and FGFR1—expressed on macrophages, have a high affinity for FGF20 [116]. Recently, some studies have reported that β-catenin is related to the polarization of M2 TAMs. Zhao et al. found that MSR1 promoted the osteogenic differentiation of BMSCs and facilitated M2-like polarization by enhancing mitochondrial oxidative phosphorylation via PI3K/AKT/GSK3β/β-catenin signaling. Yang et al. reported that Wnt ligands stimulate M2 TAMs via canonical Wnt/β-catenin signaling during the progression of HCC [117,118,119,120]. Moreover, exosomal-miR-590-3p derived from M2 TAMs promotes epithelial repair and reduces damage via the LATS1/YAP/β-catenin signaling axis [118]. β-catenin also mediates the activation of FOS-like antigen 2 (FOSL2) and repression of the AT-rich interaction domain 5A (ARID5A) driving TAMs to switch from M1-like to M2-like in lung cancer [121]. In a study by Matias et al., with the stimulation of Wnt3a, GBM and microglial cells activated the Wnt/β-catenin signaling pathway to promote M2 TAM polarization [122].

A study by Cai et al. confirmed that glucocorticoids (GCs) interact with GR on glioma upregulating gene transcription, thereby promoting the expression of FGF20. They also found that FGF20 binding with FGFR1-phosphorylated GSK3β increased the stability of β-catenin. Additionally, phosphorylated GSK3β did not influence the β-catenin translational level but inhibited its degradation by reducing the ubiquitination level. FGF20 also promotes β-catenin entry into the nucleus and execution of transcriptional functions. Then, stabilized β-catenin promotes glioblastoma cell migration and invasion, as well as M2 TAM polarization [123]. It was also reported that the β-catenin/TCF/LEF complex binding to the CD274 gene promoter region induced by the Wnt ligand and activated EGFR promoted PD-L1 expression, which promotes glioblastoma immune evasion [124]. Upregulated N-cadherin induced by radiation leads to the accumulation of β-catenin at the GBM cell surface, suppressing Wnt/β-catenin proliferative signaling, which reduces neural differentiation and protects against apoptosis [125]. As mentioned previously, the core regions of GBM are more hypoxic and acidic than the peripheral regions, and hypoxia-inducible factor (HIF)-1α activated by hypoxia via inhibition of HIF-1α prolyl hydroxylation, or by β-catenin/T-cell factor 4 complex binding with STAT3, upregulates the canonical Wnt/β-catenin pathway, which promotes proliferation, invasion, apoptosis, vasculogenesis, and angiogenesis [126]. According to a study by Yin et al., arsenite-resistance protein 2 (ARS2)—a zinc finger protein—directly activates its novel transcriptional target MGLL, which encodes monoacylglycerol lipase (MAGL). MAGL can hydrolyze endocannabinoid 2-arachidonoylglycerol (2-AG) to arachidonic acid (AA), which can be enzymatically converted to prostaglandin E2 (PGE2). PGE2 stimulates β-catenin accumulation and activation to regulate GSC self-renewal via the phosphorylation of LRP6 and promotes M2 TAM polarization [127].

### 2.9. The JMJD1C/mir-302a/H3K9/METTL3/SOCS2/M1 Pathway

DNA or RNA methylation and demethylation as well as histone methylation and deacetylation have been confirmed to be closely related to tumor development. As previously reported, m6A demethylase ALKBH5 demethylates FOXM1, leading to enhanced FOXM1 expression, thereby preserving the tumorigenesis of GBM stem-like cells [128]. The evidence that histone deacetylases (HDACs) are linked to macrophage inflammatory pathways and anti-bacterial responses, macrophage metabolism, and myeloid development, provides the rationale for their use in cancer treatment, inflammatory, and infectious diseases [129]. Another study demonstrated that the histone demethylase JMJD3 regulates the transcription of M2-associated genes, such as Arg1, Chi3l3 (Ym1), and Retnla (Fizz1) through the methylation of histone H3 Lys4 (H3K4) and Lys27 (H3K27). Further, the upregulation of JMJD3 induced by IL-4 also regulates M2 macrophage polarization by inducing the expression of the transcription factor IRF4 [130,131].

Jumonji domain-containing 1C (JMJD1C), which removes demethylated histone H3 Lys9 (H3Lys9me2) via binding to the miR-302 promoter, plays a role in several tumors [132,133]. JMJD1C regulates aberrant metabolic processes in acute myeloid leukemia, contributes to MLL-AF9/HOXA9-mediated self-renewal of leukemia stem cells, and suppresses leukemia cell growth by catalyzing H3K9 demethylation [134,135,136]. JMJD1C also reportedly demethylates STAT3 to restrain plasma cell differentiation and rheumatoid arthritis to participate in the progression of prostate cancer via an upregulation of TNF-α and mediate myocardial hypertrophy via angiotensin II (Ang II) [133,137,138]. In GBM cells, when JMJD1C was overexpressed, Zhong et al. found a decrease in CD206^+^ cells and an increase in CD86^+^ cells in tumor tissues. Additionally, the levels of M1 markers such as IL-1β, TNF, CXCL9, IL-23, ROS1, IL-12a, and IL-12b were increased, while the level of M2 markers decreased. These results suggest that JMJD1C promotes M1 macrophage polarization. Furthermore, they found that JMJD1C in GBM cells upregulates miR-302a by promoting H3K9 demethylation at the promoter region, and then miR-302a targets the methyltransferase-like 3 (METTL3) 3′UTR to negatively regulate its expression. Additionally, METTL3 promotes the suppressor of cytokine signaling 2 (SOCS2) degradation in gliomas by promoting an m6A methylation modification of SOCS2 to inhibit M1 macrophage polarization. Thus, JMJD1C promotes M1 macrophage polarization via the miR-302a/METTL3/SOCS2 axis [139] (Figure 2).

Recent studies have shown that METTL3 recognized by YTHDF2 mediates m^6^A modification to activate NF-κB and promote the malignant progression of glioma. Tassinari et al. found that the upregulation of METTL3 in GBM cells methylates ADAR1 mRNA, leading to a pro-tumorigenic mechanism connecting METTL3, YTHDF1, and ADAR1. Additionally, METTL3 upregulates the expression of COL4A1 by reducing its methylation level to participate in glioma development. Furthermore, the overexpression of METTL3 also suppresses GSC growth and self-renewal [140,141,142,143].

In addition to SOCS2, SOCS1 also has a pro-M1 polarization effect on macrophages. The SOCS family (CIS and SOCS1-7) with a variable amino-terminal region, central Src homology (SH2) domain, and conserved carboxyl-terminal domain (SOCS-box) inhibits the JAK/STAT pathway [144]. The SH-domain of SOCS1 interacts with JAK2 to suppress its phosphorylation, thereby inhibiting STAT3 [145]. Reportedly, the SOCS1 is lowly expressed or methylated and silenced in tumors of the lung, liver, colon, and head and neck, as well as in GBM [146]. DNA methylation, involving the transfer of a methyl group to the 50-C in CpG dinucleotides via DNMT1 (responsible for maintenance of methylation patterns onto daughter strands), DNMT3A, and DNMT3B (collectively responsible for de novo methylation), participates in cell cycle regulation, DNA repair, and GBM progression [147,148]. Some studies demonstrated that the inhibition of DNMT suppressed DNA repair activity to enhance the radiosensitivity of human cancer cells and that the inhibition of DNMT3A/ISGF3γ interaction increased the efficiency of temozolomide to reduce tumor growth [149,150]. Furthermore, DNMT1 methylates the CpG island of the SOCS1 promoter and coding sequence, leading to the loss of SOCS1 expression and production of TNF-α and IL-6, which promotes the polarization of M1 TAMs [145,151]. DNMT3B, similar to DNMT1, directly binds to the proximal promoter and 5′-untranslated regions of PPARγ1 in macrophages, leading to DNA methylation, and the inhibition of DNMT3B can lead to M2 TAM polarization [152].

### 2.10. The (Gal3BP/Gal)/CHI3L1/M2 Pathway

Chitinase-3-like protein-1 (CHI3L1), known as human homolog YKL-40, belonging to the glycoside hydrolase family, is highly expressed in various tumors, especially in GBM, and is involved in extracellular tissue remodeling, Th1/Th2 inflammation, oxidative injury, apoptosis, pyroptosis, angiogenesis, and parenchymal scarring [153,154,155]. CHI3L1 is highly expressed in glioblastoma and is related to poor patient prognosis. It modulates adhesion, rearranges the actin cytoskeleton, and expresses MMP-2 to regulate glioma cell invasion [156,157]. It has also been reported that CHI3L1 promotes glioma progression via the NF-κB signaling pathway and reprograms the tumor microenvironment [153].

Chen et al. found that CHI3L1, regulated by the PI3K/AKT/mTOR pathway in a positive feedback loop and in a time- and dose-dependent manner, regulates TAM polarization toward the M2-like phenotype in the GBM TME. Galectin 3 (Gal3) binding to CHI3L1 activates the AKT/mTOR-mediated transcriptional regulatory network (NF-κB and CEBPβ), leading to an immune suppression and polarization of M2 TAMs, which then activate PD-1 and CTLA-4 to promote tumor progression (Figure 2). Additionally, galectin 3-binding protein (Gal3BP), also known as 90K or Mac-2 binding protein, encoded by the LGALS3BP gene, competes with Gal3 to bind with CHI3L1 and negatively regulates M2-like macrophage migration and promotes M1-like TAMs [158].

### 2.11. The MFG-E8/ITGB3/STAT3/M2 Pathway

Milk fat globule-epidermal growth factor 8 (MFG-E8), also known as lactadherin, widely participates in the phagocytic clearance of apoptotic cells, inhibition of inflammation, and reversal of cellular oxidative stress, and is also involved in several tumors such as melanoma, breast cancer, and colorectal cancer [159,160,161,162,163]. Reportedly, MFG-E8 binds with αvβ3/αvβ5-integrin via the RGD motif, then recognizes and tightly binds to phosphatidylserine via the C1 and C2 domains of apoptotic cells to promote phagocytic engulfment [164]. Furthermore, in prostate cancer, it facilitates M2 polarization via the SOCS3/STAT3 pathway [165]. It is reportedly suppressed by Connexin43 to induce contact growth inhibition in glioma cells [166].

In a study by Wu et al., MFG-E8 was significantly upregulated in glioma. They also reported that MFG-E8 binding with its receptor integrin β3 (ITGB3) in an autocrine or paracrine manner mediates the activation of STAT3 to promote the polarization of M2 TAMs, thereby secreting factors such as TGF-β, VEGF, IL-10, ARG-1, MGL2, and CD206 to promote glioma progression (Figure 3). Furthermore, they found that the knockdown of MFG-E8 inhibits glioma cell growth via the ITGB3/FAK/ERK or ITGB3/STAT3/cyclin D3 signaling pathways and suppresses M2 polarization [167].

### 2.12. The Exosome Pathway

Exosomes (containing peptides, proteins, mRNA, miRMA, and lncRNAs) derived from multivesicular bodies (MVBs) are small bilayer-membranous extracellular vesicles between 40 and 150 mm in diameter, and they play an important role in tumor angiogenesis, immune suppression, tumor invasion, and treatment resistance. Recently, several studies have confirmed that exosomes can transfer bioactive molecules to mediate tumorigenesis and angiogenesis of GBM, and they can also regulate the polarization of TAMs [5,6,27,168]. Recent research has shown that PD-L1 on GSC-EVs can prevent the activation and proliferation of CD8^+^ T cells, and that it binds with PD1, helping GBM evade T-cell infiltration in the TME [5,169,170,171]. In medulloblastoma (MB), downregulation of let-7i-5p and miR-221-3p derived from SHH MB-Exo promotes the polarization of M2 TAMs via the upregulation of PPAR-γ [168]. In GBM, the transfer of miR-21 and miR-451 mediated by GBM-derived extracellular vesicles (GDEV) leads to the downregulation of c-Myc expression as well as its target gene BTG2, followed by the polarization of M2 TAM [169]. It has been reported that miR-1246 targets telomeric repeat-binding factor 2-interacting protein 1 (TERF2IP) to activate the STAT3 pathway and inhibit the NF-κB pathway, inducing M2 polarization [172], while miR-124 suppresses STAT3 signaling to inhibit M2 polarization. STAT3 phosphorylated by JAK dimerizes and migrates to the nucleus, and modulates proliferation, migration, and inflammation. MiR-124 inhibits the phosphorylation of Tyr705 and Ser727 on STAT3, thereby disrupting the balance of STAT1/STAT3 [173]. Otherwise, miR-1246 induced by hypoxia has also been reported to promote M2 polarization by targeting telomeric repeat-binding factor 2-interacting protein 1 (TERF2IP), thereby activating STAT3 and inhibiting the NF-κB pathway [169]. In a study by Pan et al., circNEIL3—a special class of noncoding RNAs without a 5′end cap or a 3′ end poly(A) tail—cyclized by EWS RNA-binding protein 1 (EWSR1), is positively related to the malignant progression of glioma. They confirmed that circNEIL3 packaged into exosomes by hnRNPA2B1 was transmitted to TAMs, stabilizing insulin-like growth factor 2 mRNA binding protein 3 (IGF2BP3) by preventing HECTD4-mediated ubiquitination to increase YAP1 expression, which promotes M2 polarization [6]. MiR-155 decreases the expression level of IL-13 receptor α1 (IL13Rα1), inhibiting the activation of STAT6, thus regulating the expression of M2-related genes, such as CD23, DC-SIGN, CCL18, and SERPINE in an IL13-dependent manner [32,174]. M1 polarization is also supported by miR-127 and miR-124, and miR-127 inhibits Bcl6, leading to downregulation of Dusp1 and upregulation of JNK phosphorylation, thereby promoting M1 polarization [175]. Other studies have also confirmed that miR-511-3p, miR-22, miR-99, miR-32, and miR-142-3p promote M2 polarization, while miR-155, miR-181, miR-451, and miR-504 promote M1 polarization; however, the exact mechanism remains to be investigated [151,176,177,178].

### 2.13. Other Pathways

In addition to the pathways mentioned above, several other pathways also contribute to the polarization of M2 macrophages. The evidence from a study by Xu et al. has confirmed that immunity-related GTPase M (IRGM) as a member of the GTPases family is highly expressed in glioma, and it significantly increases the expression of P62, necrosis factor receptor activating factor 6 (TRAF6), and NF-kB transportation to the nucleus. The interaction of P62 with TRAF6 leads to TRAF6 autoubiquitination and NF-kB activation, which mediates lL-8 production to promote M2 polarization and macrophage inflammation protein 3-α (MIP-3α) secretion to recruit macrophages [179]. According to a study by Zhang et al., Class A1 scavenger receptor (SR-A1), a pattern recognition receptor, is mainly expressed in macrophages or microglia in the brain, and its expression in gliomas is positively correlated with tumor grade and negatively correlated with patient prognosis. The deletion of SR-A1 promotes the polarization of M2 TAMs via the activation of STAT3 and STAT6. HSP70, an endogenous ligand of SR-A1, induces the inhibition of STAT3 and STAT6 to promote M1 TAM polarization, thereby suppressing the progression of glioma [180]. Another study demonstrated that CXC motif chemokine ligand 13 (CXCL13), a CXC chemokine specifically binding to CXC chemokine receptor type 5 (CXCR5) to prolong the activation of oncogenic kinases and signaling, is linked to CD163^+^ M2 TAMs and could also promote the tumor polarization of M2c via IL-10 induction [8].

## 3. The Effect of M1 and M2 TAM on Glioma Progression

As mentioned above, TAMs in glioma are divided into tumor-suppressive M1- and M2-like subtypes. The former inhibits the progression of glioma, while the latter has the opposite effect. M2 TAMs secrete a variety of inflammatory factors and chemokines—such as TGF-β, IL-10, VEGF, MMP, CCL15, CCL17, and CCL22—to stimulate angiogenesis, maintain tumor cell stemness, facilitate immune infiltration, remodel tissue, and induce drug resistance, thus promoting M2 polarization [10,48,181]. M1 TAM highly expressing TNF-α, IL-1β, IL-6, IL-8, IL-12, and IL-23 promote strong T helper 1 (Th1) responses and activate natural killer (NK) cells [10]. Reportedly, GBM-derived pro-inflammatory factors such as IL-4, IL-10, IL-13, CCL2, and macrophage colony-stimulating factor (M-CSF) promote the M2 phenotype of macrophages [13,48]. M2 macrophages secrete a large amount of TGF-β, which binds to a dimeric receptor complex of TGF-βI receptor (TGF-βRI) and TGFβRII, leading to SMAD phosphorylation or initiation of non-SMAD signaling [182,183]. Reportedly, these receptors, including TGFβRI and TGFβRII, are endowed with intrinsic serine/threonine kinase activity [184]. M2 macrophages have also been reported to secrete TGF-β, inducing the transfer of immature CD4^+^ T to Treg cells and promote their proliferation [185].

TGF-β, which is activated in an integrin-dependent manner, exists in latent forms through binding to the extracellular matrix to form large complexes or binding to the glycoprotein-A repetitions predominant (GARP) transmembrane domain [181,186]. It functions by immune suppression, migration, invasiveness, angiogenesis, and maintenance of the stemness of GBM stem cells (GSCs) [181,187]. TGF-β has three isoforms—TGF-β1, TGF-β2, and TGF-β3—each of which is synthesized as a homodimer, interacting with latency-associated peptide (LAP) and latent TGFβ-binding protein (LTBP) to form the large latent complex (LLC). Additionally, LLC is released from the ECM, then LAP is further hydrolyzed to release active TGF-β to its receptor, which comprises the activation process of TGF-β [188].

The activated dimeric TGF-β ligand interacts with cell-surface transmembrane receptors (TGFβRII), which recruit and phosphorylate TGF-β type I receptors (TGFβRI), and then activate TGFβRI, which phosphorylates SMAD2 and SMAD3 at the C-terminal serine residues. Phosphorylated SMAD2 and SMAD3 can assemble into heterodimeric and trimeric complexes with the common mediator SMAD4 and then translocate to the nucleus, where they regulate transcriptional responses [183,188,189]. After entering the nucleus, the heteromeric complexes (SMAD2/3–SMAD4) combine with the genomic SMAD-binding element (SBE) in a sequence-specific manner to recognize target genes and regulate transcription. Once in the nucleus, SMAD3 and SMAD4 bind directly to DNA, whereas SMAD2 is responsible for splicing. SMAD7, an inhibitor of the TGF-β pathway, mediates the degradation of the type I receptor, inhibits the phosphorylation of SMAD2/3, and suppresses the formation of the SMAD2/3–SMAD4 complex [183,189,190,191]. As for non-SMAD signaling, some studies noted that TGFβRI directly activates the PI3K/AKT/mTOR/S6K pathway to control translation, the RHO/ROCK/LIMK/cofilin pathway to change the actin cytoskeleton, the TRAF(4/6)/TAK1/MKK/(P38 or JNK) pathway to regulate transcription, and the PAR6/SMURF/RHO pathway to resolve tight junctions. Additionally, activated TGFβRI also phosphorylates SRC homology domain 2-containing protein (SHC) to recruit the proteins GRB2 and SOS, followed by activation of ERK MAPK signaling (RAS/RAF/MEK/ERK) [183,188,192]. Furthermore, another study reported that sex-determining region Y-box 4/2 (SOX4/2), including SOX2 and SOX4, induced by the SMAD2/3 pathway, which is activated by TGFβ1 (mainly from M2 TAM), promotes the stemness and migration abilities of glioma cells [190].

An extracellular secreted matrix protein, transforming growth factor beta-induced (TGFBI, originally named βig-h3), comprises 683 amino acids that have been shown to play a critical role in morphogenesis, differentiation, inflammation, tumor progression and metastasis and cell growth [24,193,194]. TGFBI is thought to play an important role in cancer, and Lang et al. found that TGFBI is a new urinary biomarker for muscle-invasive and high-grade urothelial carcinoma (UC) that participates in the proliferation and migration of cancerous urothelial cells [194]. Ween et al. found that TGFBI has dual functions, acting both as a tumor suppressor and tumor promoter [195]. Peng et al. found that TGFBI, preferentially secreted by M2-like TAMs in the glioma microenvironment, was negatively associated with overall survival and mediated the pro-tumorigenic effect of M2-like TAMs in GBMs [24].

Arg-Gly-Asp (RGD) motifs contained within the protein structure of TGFBI, called integrin-binding motifs, were shown to bind to integrins located on the cellular surface in tumors, including α1β1, α3β1, α5β1, α6β1, αvβ1 αvβ3, αvβ5, and αvβ6 [188,195,196,197,198]. TGFBI secreted by M2-like TAMs binds to integrin ανβ5, activating the phosphorylation of the tyrosine kinase Src, leading to the upregulation of the STAT3 pathway, which promotes the progression of GBM [24]. Furthermore, activated STAT3, which is associated with a poor prognosis in cancer, participates in tumor progression via IL-10 and IL-6, and tightly regulates M2 polarization and activation status [18,199].

Integrin is also involved in a number of GBM mechanisms. Periostin generated by GSCs, accruing in the perivascular niche, induces integrin αvβ3 receptor signaling to recruit TAMs (M2-like), maintain the microglia or macrophages M2 phenotype and promote extravasation and migration in the glioma environment [200,201,202]. It has also been found that M2 TAMs upregulate the expression of TGFβ-1, which increases the level of high mobility group AT-hook 2 (HMGA-2), and then HMGA-2 inhibits the expression of miR-340-5p. Downregulation of miR-340-5p promotes TAM recruitment by periostin-mediated αvβ3 integrin signaling and regulates M2-macrophage polarization by TGF-binding protein (LTBP-1)-mediated TGFβ-1 [203]. Additionally, osteopontin (OPN), which maintains the M2 macrophage gene signature and phenotype, is derived from tumor cells and macrophages. OPN interacts with its receptor, αvβ5 integrin, which is highly expressed in macrophages of GBM, to recruit macrophages [198]. WISP1 secreted by GSCs regulates the integrin α6β1-AKT pathway to promote the survival of M2 TAMs [204].

NOS2, a heme-containing enzyme that catalyzes the synthesis of NO and citrulline from L-Arg, is expressed by M1 macrophages, whereas M2 macrophages express arginase I and II—manganese metalloenzymes that metabolize arginine into urea and L-ornithine, respectively. The expression of NOS2 can be activated by NF-κB, JAK3, STAT1, and JNK and is upregulated by proinflammatory cytokines, bacterial LPS, and hypoxia. ARG1 is induced by TGF-β, macrophage-stimulating protein (MSP), GM-CSF, IL-4, and IL-13 in the cells of the innate immune system. Otherwise, L-Arg metabolism has been reported to impair the antigen responsiveness of T cells at the tumor–host interface [12,35,205]. NO produced by iNOS or NOS2 in glioma displays cytoprotective properties at low physiological levels, while producing toxic effects at high concentrations at high levels. NO induces the phosphorylation of syntaxin 4 (synt4) via the generation of cyclic GMP and activation of protein kinase G (PKG) [206]. Synt4 is a SNARE protein responsible for A-SMase trafficking and activation, and it is necessary for A-SMase plasma membrane localization and translocation, which is essential for initiating apoptosis [207]. Interestingly, PKG phosphoglycerates synt4 at serine 78 leads to proteasome-dependent degradation of synt4, which inhibits A-SMase-dependent apoptosis and stimulates cell survival and proliferation [206,207].

Zhang et al. showed that M2 macrophages enhance 3-phosphoinositide-dependent protein kinase 1 (PDPK1)-mediated phosphoglycerate kinase 1 (PGK1) threonine (T) 243 phosphorylation in tumor cells via the secretion of IL-6, which facilitates a PGK1-catalyzed reaction toward glycolysis by altering substrate affinity. PGK1 T243 phosphorylation mediates macrophage-promoted glycolysis, tumor cell proliferation, and gliomagenesis and correlates with macrophage infiltration, grades, and the prognosis of GBM patients [208]. Zhu et al. showed that Cat Eye Syndrome Critical Region Protein 1 (CECR1), a member of the adenyl-deaminase growth factor family, which is highly expressed in M2 TAMs in GBM, regulates the expression of PDGFB, promotes angiogenesis and pericyte migration, and enhances the expression and deposition of periostin via PDGFB-PDGFRβ signaling. Additionally, pericytes are closely involved in vascular construction, maintenance, regulation of vascular physiology, and immune cells recruitment [209,210]. Periostin, a downstream target of PDGFB signaling in pericytes, is known as a proangiogenic extracellular matrix component in glioma [211], and plays an important role in regulating cell migration and epithelial–mesenchymal transition through binding with integrins to activate cell focal adhesion kinases; it is also involved in angiogenesis, the polarization of M2 TAMs, and tumor progression [201,212,213].

Moreover, Zhang et al. found that M2 TAMs in gliomas drive vasculogenic mimicry by amplifying IL-6 secretion in glioma cells via the PKC pathway [214]. A study by Qi et al. showed that IL-10 from glioma could form a complex with JAK2, activating the JAK2/STAT3 pathway to promote tumorigenesis [215]. Lastly, exosomes derived from M2 TAMs have been shown to promote glioma progression in several studies. CircKIF18A from glioblastoma-associated microglia can bind to FOXC2 in human brain microvessel endothelial cells (hBMECs), and maintain the stability and nuclear translocation of FOXC2. FOXC2, a transcription factor, binds to and upregulates the promoters of ITGB3, CXCR4, and DLL4, which activate the PI3K/AKT/mTOR signaling pathway to promote the growth and angiogenesis of tumors [27]. MicroRNA-155-3p from M2 macrophage-derived exosomes directly targets WDR82 in the MB to decrease its expression, thereby promoting the invasion, growth, and migration abilities of MB cells [216]. Yao et al. found that miR-15a and miR-92a were lowly expressed in M2 macrophage-derived exosomes, decreased the expression of their target genes CCND1 and RAPIB, blocking the PI3K/AKT/MTOR signaling pathway to inhibit the migration and invasion of glioma cells [217]. It has also been reported that miR-7239-3p upregulated in M2 microglial exosomes is recruited to glioma cells and inhibits Bmal1 expression, thereby promoting the progression of glioma [218].

## 4. The Treatments for Gliomas

In clinical practice, surgery is a common treatment for glioma, especially glioblastoma, and postoperative adjuvant radiotherapy and chemotherapy can improve the prognosis of patients to a certain extent. In addition, many experts have proposed new treatment schemes based onto the various glioma mechanisms. Although most of these are still not applied in clinical practice, they provide a new reference for clinical treatment. As is previously mentioned, TAMs can be functionally divided into M1 (suppressing glioma) and M2 (promoting glioma) and, currently, therapeutic strategies targeting the polarization of M1 and M2 macrophages have emerged. The current therapeutic direction is mainly to reverse the polarization of M2 TAMs and reprogram them into M1 TAMs. Enhancing M1 activation by targeting the CD47-SIRPα signaling pathway and targeting the CD40 protein will reinforce the cytotoxic/phagocytic potential of TAM, which mediates antibody-dependent cellular cytotoxicity/phagocytosis [18]. CD47, an immunoglobulin, interacts with SIRPα on macrophages, leading to the phosphorylation of the cytoplasmic part of the SIRPα immunoreceptor tyrosine-based inhibition motif (ITIM), which, as an immune checkpoint, transmits an inhibitory signal [219,220,221]. Phosphorylated ITIM leads to the accumulation of deactivated myosin IIA, which inhibits M1 cells from engulfing tumor cells [222]; thus, targeting the CD47–SIRPα axis enhances tumor cell M1-mediated phagocytosis [219,223,224]. Meanwhile, according to a study by Heidari et al., anti-CD47 treatments were shown to enhance tumor-cell phagocytosis by M1 and M2 macrophages, with a higher phagocytosis rate by M1 macrophages [225]. Ma et al. knocked out CD47 and found that more M2-like TAMs were recruited to induce phagocytosis and decrease intracranial tumor growth [226]. CD40, another target of M1 activation and member of the tumor necrosis factor receptor family, activates APC to promote CD8+ T cell responses and induce tumor collapse [18,227]. Thus, CD40 agonists may be a promising future GBM therapy. Recently, immunotherapy has become a hot topic in the treatment of gliomas, and the discovery of immune checkpoint inhibitors, chimeric antigen receptor therapy, and dendritic cell vaccines has provided clinicians with new treatment strategies [181]. PD-1, the most well-known checkpoint inhibitor expressed by activated T cells, can interact with PD-L1 to downregulate the T-cell receptor (TCR) and CD signaling and promote the vitality, growth, proliferation, and migration of GBM [5,181,228]. Interestingly, it has recently been shown that a high expression of PD-1 was significantly related to M2-polarization of macrophages (M2-TAMs), and anti-PD-1 treatment could reverse the transition of TAMs from M1 to M2 [1,229,230]. However, according to another study, although blocking PD-1 increases T cell infiltration and trafficking, as well as immunosuppressive activity, it is unable to overcome this M2 macrophage polarization [181]. Additionally, CpG-decorated gold (Au) nanoparticles (CpG@Au NPs) were applied to improve the RT/ICB efficacy by immune modulation under low-dose X-ray exposure, where Au NPs acted as nanocarriers to deliver a Toll-like receptor 9 agonist (CpG) to reprogram M2 to M1 TAMs, thus arousing innate immunity and priming T cell activation [231]. Meanwhile, various treatments have been developed for the various mechanisms of macrophage polarization in glioblastoma, such as the receptor CD74 on GAMs activated by macrophage migration inhibitory factor (MIF), which is secreted by brain tumors, escapes pro-inflammatory M1 conversion and promotes the M2 shift of microglial cells. MIF-CD74 signaling phosphorylates microglial ERK1/2, inhibiting interferon (IFN)-γ secretion in microglia, thereby promoting tumor growth and proliferation, and treatments toward inhibiting CD74 or MIF will lead to tumor death [232,233]. A CMKLR1 inhibitor, α-NETA, blocks the chemerin/CMKLR1 axis, reduces TAM infiltration as well as NF-κB signaling, and upregulates the anti-tumor functions of T cells [61]. Celastrol (CELA), as an anti-tumor compound, reportedly promotes M1 polarization by inhibiting the p-STAT6 signaling pathway and downregulate the TGF-β1 level to inhibit the tumor progression, and was shown to inhibit M2-like polarization of macrophages. This indicates the potential application of celastrol in GBM treatment. Zhu et al. designed a biomimetic BBB-penetrating albumin nanosystem modified by a brain-targeting peptide to help celastrol cross the blood–brain barrier and enter the glioma [9,182]. Astrocyte elevated gene-1 (AEG-1) overexpressed in glioma interacting with GSK-3β activates Wnt/β- catenin signaling, and a knockdown of AEG-1 sensitizes glioma cells to TMZ, promotes TMZ-induced DNA damage, and decreases M2-polarization [234]. Therefore, targeting AEG-1 may be a promising treatment for improving the efficiency of chemotherapy and reducing M2 polarization. Another study reported that via integrin α6β1-AKT signaling, Wnt-induced signaling protein 1 (WISP1), which is expressed and secreted by GSCs, maintains GSCs and M2 TAMs, respectively, by an autocrine mechanism and in a paracrine manner; thus, targeting Wnt/βcatenin-WISP1 signaling might be an effective GBM therapy and may provide a reference for future treatments [204]. Li et al. found that palbociclib modulates the lncRNA SNHG15/CDK6/miR-627 circuit, which overcomes temozolomide resistance and reduces M2-polarization of glioma [235]. Additionally, many drugs such as ocoxin, chlorogenic acid, dopamine, oleanolic acid, and corosolic acid also inhibit M2 polarization in gliomas [236,237,238,239,240] (Figure 4).

## 5. Conclusions

Glioma is the most common and malignant tumor of the central nervous system; however, even including surgical treatment, which is only viable for some patients, there are currently few effective treatments for glioma, especially glioblastoma. Therefore, the mechanisms involved in the occurrence, development, and metastasis of GBM will provide a reference for future research on glioma treatment, improving the prognosis of patients and increasing their survival rate.

## Figures and Tables

**Figure 1 brainsci-13-01269-f001:**
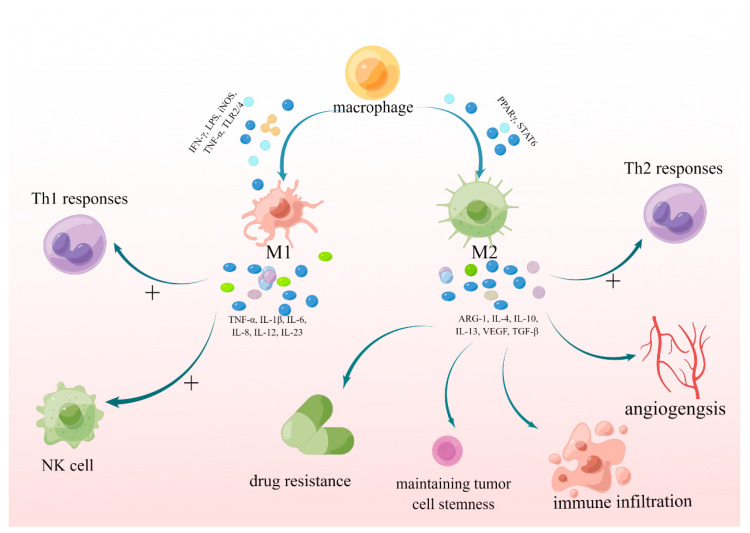
The polarization of macrophages and the functions of M1 and M2 TAMs. TAMs can be divided into tumor-suppressive M1-like and tumor-supportive M2-like polarized cells. M1 TAMs can secret TNF-α, IL-1β, IL-6, IL-8, IL-12, and IL-23 promoting T helper 1 (Th1) responses, and also activate natural killer (NK) cells. M2 TAMs can activate Th2-type immune response, stimulate angiogenesis, maintain tumor cell stemness, facilitate immune infiltration, and induce drug resistance by secreting ARG1, IL-13, IL-10, IL-4, VEGF, and TGF-β1. ARG-1: arginase 1; IFN-γ: interferon gamma; IL: interleukin; iNOS: inducible nitric oxide synthase; LPS: lipopolysaccharides; NK: natural killer; PPARγ: peroxisome proliferator-activated receptor-γ; STAT: signal transducer and activator of transcription 3; TAMs: tumor-associated macrophages; TGF-β: transforming growth factor-β; Th1: T helper 1; TLR2/4: Toll Like Receptor2/4; TNF-α: tumor necrosis factor; VEGF: vascular endothelial growth factor.

**Figure 2 brainsci-13-01269-f002:**
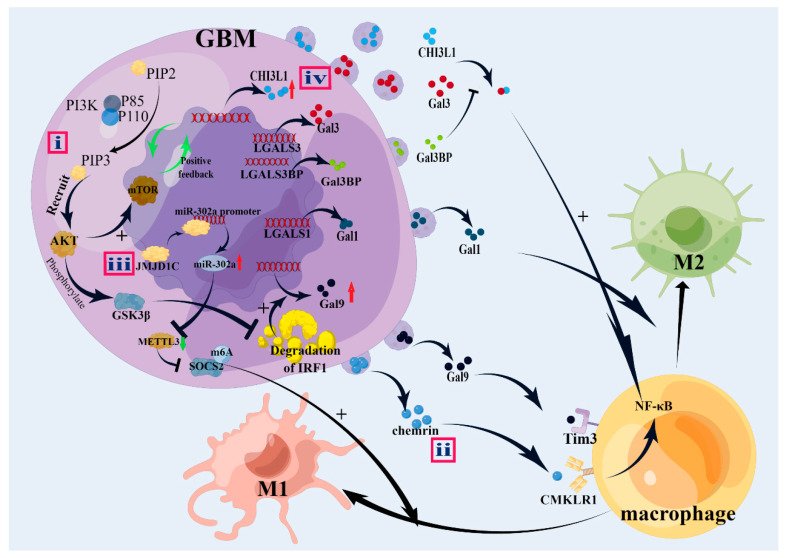
Partial mechanisms of M1 and M2 TAMs polarization. i: PTEN-deficient GBM cells activate PI3K/AKT pathway, which suppresses IRF1 degradation by phosphorylating glycogen synthase kinase 3 (GSK3β) into an inactive form. Then, upregulated Gal-9 interacts with Tim3 to promote the polarization of M2 TAMs. ii: Chemerin binding with CMKLR1 activates NF-κB pathway, which promotes M2 polarization. iii: JMJD1C upregulates miR-302a by promoting H3K9 demethylation at the promoter region, then negatively regulating the expression of METTL3. Therefore, downregulated METTL3 inhibits SOCS2 degradation by inhibiting m6A methylation modification to promote M1 polarization. iv: CHI3L1 regulated by the PI3K/AKT/mTOR pathway in a positive feedback loop binds with Gal3-activating NF-κB pathway to promote M2 polarization. AKT: protein kinase B; CHI3L1: chitinase-3-like protein-1; CMKLR1: chemokine-like receptor 1; Gal: galectin; Gal3BP: galectin 3-binding protein; GBM: glioblastoma; GSK3β: glycogen synthase kinase 3 beta; IRF1: interferon regulatory factor 1; JMJD1C: jumonji domain-containing 1C; NF-κB: nuclear factor kappa-B; METTL3: methyltransferase-like 3; mTOR: mammalian target of rapamycin; PIP2: phosphatidylinositol (4,5)-bisphosphate; PIP3: phosphatidylinositol (3,4,5)-triphosphate; P85: regulatory subunit; P110: catalytic subunit; PI3K: phosphatidylinositol 3 kinase; SOCS2: suppressor of cytokine signaling 2; m6A:; Tim3: T cell immunoglobulin and mucin domain 3. Meaning of symbols: +, promotion or activation; -----|, inhibition; ↑, upregulated.

**Figure 3 brainsci-13-01269-f003:**
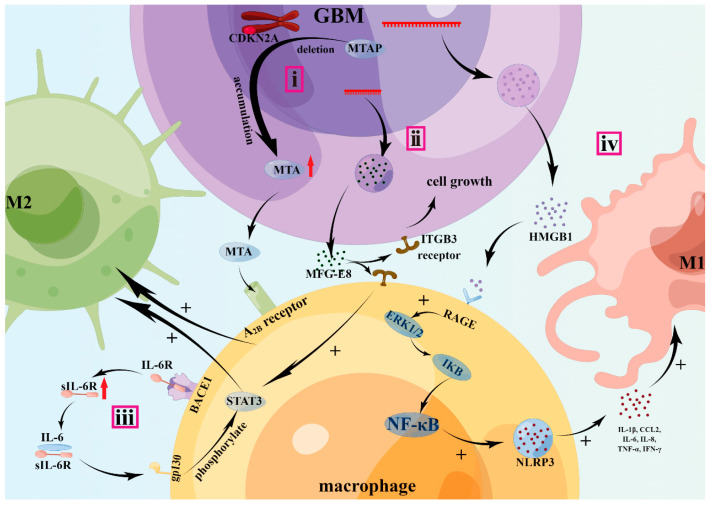
Molecular mechanisms of M1 and M1 TAMs polarization. i: The deletion of MTAP often leads to the accumulation of MTA, which binds with A_2B_ receptors to promote M2 polarization. ii: MFG-E8 binding with ITGB3 not only promotes GBM growth in an autocrine manner, but also promotes M2 polarization via activation of STAT3. iii: BACE1 acts as a transmembrane protease mediating the shedding of IL-6R, and soluble IL-6 receptor (sIL-6R) in extracellular matrix binds with IL-6 forming an IL-6/sIL-6R complex. Then, the complex binds to gp130 activating phosphorylation of STAT3 to promote M2 polarization. iv: HMGB1 activated RAGE through the phosphorylation of ERK1/2 and IKB, then activating the RAGE/NF-κB/NLRP3 inflammasome pathway, which promoted the release of TNF-α, IFN-γ, IL-1β, IL-6, IL-8, and CCL2, thereby enhancing M1-like polarization of TAMs. BACE1: β-site AβPP-cleaving enzyme 1; CDKN2A: cyclin-dependent kinase inhibitor 2A; GBM: glioblastoma; gp130: glycoprotein 130; HMGB1: high mobility group box 1 protein; ITGB3: integrin β3; MFG-E8: milk fat globule-epidermal growth factor 8; MTA: methylthioadenosine; MTAP: methylthioadenosine phosphorylase; NLRP3: NOD-like receptor thermal protein domain-associated protein 3; RAGE: receptor for advanced glycation end products; sIL-6R: soluble IL-6 receptor; STAT: signal transducer and activator of transcription 3. Meaning of symbols: +, promotion or activation; ↑, upregulated.

**Figure 4 brainsci-13-01269-f004:**
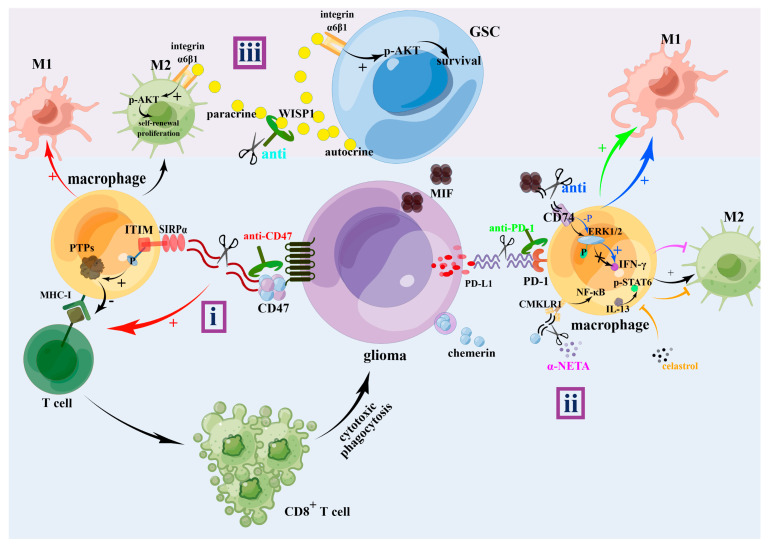
The treatments of the glioma. i: Binding of CD47 to SIRPα promotes phosphorylation of ITIM, which recruits and activates protein tyrosine phosphatases (PTPs). Anti-CD47 treatment not only will promote the polarization of M1 macrophages, but also promote cytotoxic phagocytosis of CD8+ T cell. ii: Anti-PD-1 treatment, α-NETA severing chemerin/CMKLR1/NF-κB axis, celastrol severing IL-13/p-STAT6 axis, anti-CD74 or anti-MIF severing MIF/CD74/p-ERK1/2/IFN-γ axis all promoted M1 polarization and inhibited M2 polarization. iii: WISP1 autocrined by GSC maintains GSCs through integrin α6β1-AKT. And it also promotes a self-renewal proliferation of M2 TAMs through a paracrine manner. α-NETA: 2-(alpha-naphthoyl)ethyltrimethylammonium; CD47: cluster of differentiation 47; CMKLR1: chemokine-like receptor 1; ERK1/2: extracellular regulated protein kinases; GSC: glioma stem cells; ITIM: immunoreceptor tyrosine-based inhibitory motifs; MHC-I: the major histocompatibility complex class I; MIF: migration inhibitory factor; PD-1: programmed cell death protein 1; PD-L1: programmed cell death 1 ligand 1; PTPs: protein tyrosine phosphatases; SIRPα: Signal Regulatory Protein α; WISP1: Wnt-induced signaling protein 1. Meaning of symbols: +, promotion or activation; **-----|**, inhibition.

## Data Availability

All the information relevant to the present study are available from the corresponding author on reasonable request.

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
