# Peer review of "The Importance of M1-and M2-Polarized Macrophages in Glioma and as Potential Treatment Targets"

_brainsci, 2023, doi:10.3390/brainsci13091269_

Round 1

Reviewer 1 Report

Comments and Suggestions for Authors

The importance of M1 and M2-polarized macrophages in GBM and as a potential treatment target

General comments:

This review is a valuable recompilation of a detailed and an extensive analysis about the molecular mechanisms and pathways by which GBM-associated macrophages can turn into either M1 or M2 macrophages with a tumour suppressive or tumour-supportive function respectively, therefore with a polarized function. Furthermore, they talk about this phenomenon connected to Glioma progression and treatment possibilities.

More specific comments/format:

1.      In the first image (Table of Contents Image (TOCI), on the left-hand side there are some coloured particles that promote the progression of TAMs into M1 macrophages that have no name/identification.

2.      Table 1: It would be nice to see a more detailed/explanatory table where each marker has its own reference and also its own function/contribution to the transition of either M1/M2 macrophages.

3.      Figure 2: In the figure legend there are the sections i; ii; iii; iv that have no connection/linkage in the image.

4.      The authors must pay attention when they cite into the whole text the work of other authors: they only cite the first author without referring to the rest:

a.      Line 115: Zhu should be Zhu et. al.

b.      Line 168: Wang et al.

c.      Line 180: Wu et. al.

d.      Line 215: Landon J Hansen et al.

e.      Line 221: Ludwig et. al.

f.       Line 225: Takei et.al.

g.      Line 253: Li et. al.

h.      Line 271: Wang et. al.

i.       Line 303;311: Li et. al.

j.       Line 324: Zhao et. al.

k.      Line 326: Yang et. al.

l.       Line 332: Matias et. al.

m.    Line 335: Cai et. al.

n.      Line 393: Valentina Tassinari et. al.

o.      Line 457: Wu et. al.

p.      Line 504: Wu et. al.

q.      Line 621; 638: Zhany et. al.

Reviewer 2 Report

Comments and Suggestions for Authors

In this review by Ren et al, the authors tried to conceptualize different pathways contributing for polarization of macrophages either towards M1 or M2 phenotypes within the tumor microenvironment (TME) and thus contributing for either inhibiting or favoring the growth of glioblastoma. While this attempt provides a thorough understanding of the role of macrophages in blending the TME, few concerns that the authors might consider strengthening their work.

1.     Is it true to the fact that macrophages within the tumor microenvironment attain M2 phenotype post loss of T cells and NK cells function? Moreover, what the authors mean by “along with the inhibition of lysing tumors and the functions of T cells and NK cells, tumor growth is suppressed” (Lines29-31).

2.      Basing on the authors definitions of M2d, it is giving an impression as if all four subsets of M2 exist together within M2 and each of it is unique in terms of its function. Given the scenario that they are independent subsets which gain their identity based on the stimuli received, are the M2 subsets other than M2d lack expression of M2 markers including CD206, Fizz1, dectin-1 and arginase-1?

3.      It is well demonstrated in the literature that TH2 cytokines such as IL-4, IL-13, and IL-10 prompt M2 polarization which secrete pro-angiogenic factors including adrenomedullin, angiopoietin-2 and VEGF, and produce endogenously immunosuppressive cytokines such as TGF-β and IL-10 establishing an immunosuppressive microenvironment supported by Treg functions and hampered CD8+ T cell both infiltration and activity. Thus, these sequences of events promote tumor proliferation and progression. However, the authors report that with tumor invasion and angiogenesis, TAMs transform into M2-like cells, promoting tumor progression. The authors should thoroughly look back into the literature to address the “chicken and egg” scenario.

4.      In Table 1, markers of M1 are printed in bold while M2 in regular mode.

5.      Check for mis spelt words (For ex. Marcophage in Fig 1)

6.      This review may benefit much by use of English language more precisely.

Comments on the Quality of English Language

Though the idea in compiling several different pathways contributing for macrophage polarization within the TME sounds interesting, way the story was put in terms of language is a biggest drawback. 

Reviewer 3 Report

Comments and Suggestions for Authors

Overall, this review clearly describes the scientific findings about the mechanism by which M1- and M2-polarized macrophages promote or inhibit the growth of glioblastoma and discuss the corresponding treatments. Tumor-associated macrophages (TAMs) is a promising target. According to research and trials, it proved that TAMs-targeted therapies are hoped to be applied in cancer patients in future.

In general, the structure of this research should be clear and concise. This review paper is well written and well structured. However, there are certain points that the authors need to improve, which I indicate below:

Major comments

1. For the main text. please use the template which can be downloaded from the instructions for authors section* of the Brain Sciences website.

(*) https://www.mdpi.com/journal/brainsci/instructions

2. Please consider rewriting or delete the "Main points" section on the page 1 lines 15-20

3. Please check the text again. For example, on page 6 line 215, the authors write "Landon J. Hansen found that MTAP-deficient GBM patients"; which should be "Hansen et al. [29] found that MTAP-deficient GBM patients".

Similarly on page 8 line 284, which should be “Zhai et al. also found that”

Please look for these errors throughout the text and correct them.

4. Please the authors should produce a Figure or a Table indicating the treatment for gliomas and relate it to the review study.

5. Please consider reworking the tables. Make it more understandable and orderly for the reader. References in the Tables should be indicated in an individual table or at the end of the title and not in the title.

For example: M1 TAMs markers [1,9,11,13,24-34].

Table subtitles should go vertically.

Minor comments

- Please unify the abbreviations according to the journal Brain Sciences

- Please unify the list of references according to the journal Brain Sciences.

 Please follow the example and use the template which can be downloaded from the instructions for authors section of the Brain Sciences website.

(1) https://www.mdpi.com/journal/brainsci/instructions

(2) Kasper, E.M.; Mirza, F.A.; Kaya, S.; Walker, R.; Starnoni, D.; Daniel, R.T.; Nair, R.; Lam, F.C. Surgical morbidity in relation to the surgical approach for olfactory groove meningiomas—a pooled analysis of 1016 patients and proposal of a new reporting system. Brain Sci. 2023, 13, 896. https://doi.org/10.3390/brainsci13060896

- Please unify the numbering in the brackets throughout the text following the guidelines of the journal Brain Sciences. For example: [2, 3] for [2,3]

- Page 2, line 63 “NADPH-ossidase” Is the term correct?

- In the legend of Figure 1, define what the abbreviation TAMs means.

- In the legend of Figure 2, please list the sequences indicated in the picture, so it will be more understandable to follow the idea. Indicate up-regulated or down-regulated with up or down arrows, respectively. Indicate with stop arrows in the signalling pathway when inhibition is present.

- The title of Figure 3 should be changed. Please list the sequences indicated in the picture, so it will be more understandable to follow the idea. Indicate up-regulated or down-regulated with up or down arrows, respectively. Indicate with stop arrows in the signalling pathway when inhibition is present. In the legend of Figure 3, please define abbreviations.

- In the text Figure should be in full and not abbreviated as "Fig".

For the reasons stated above, I recommend that this work be Reconsider after major revision and I hope the outcome of this specific submission will not discourage you from the submission of future manuscripts.

Comments on the Quality of English Language

Minor editing of English language required

Round 2

Reviewer 2 Report

Comments and Suggestions for Authors

Thanks for the responses.

While citing a research, the ref. number was expected to be by the end of the  sentence. Even while starting a sentence like "Chemerin is also expressed in malignant tumors such as GBM, and a study by Wu et al.[61] demonstrated that patients with GBM have high levels of chemerin expression in both the tumor and serum, which was inversely associated with patient survival", the ref number at the middle of the sentence is so what bothering and can be at the end of the sentence.

Comments on the Quality of English Language

Language can be still improved for better clarity and understanding. 

Reviewer 3 Report

Comments and Suggestions for Authors

Many thanks to the authors for considering my comments. The manuscript has been improved, but there are still some points that need to be corrected, which I indicate below:

(1) Page 1, lines 14-15: Please remove the black circle and the text that says "Table of Contents Image (TOCI):" in the file named: "brainsci-2548678-revision (clean).pdf".

(2) It is not necessary to add "Non-published Material", as this file shows the biographies of the authors. In this section you should add material related to the manuscript, for example Figures and/or Tables that have not been added in the main manuscript, I think it is not correct to add the authors' biographies.

(3) Page 1, lines 11-13: only the following should be included, affiliation is not included: * Correspondence: [email protected]

(4) In the text, references should be followed by a space when they are cited. The same applies when placing et al, this should be separated from the reference number(s) in square brackets. For example: "intracranial tumors[1]" it must be "intracranial tumors [1]" or "Zhu et al.[53] found that the mechanism of interaction" it must be "Zhu et al. [53] found that the mechanism of interaction". Please check and correct this throughout the manuscript.

(5) A list of abbreviations is not necessary, abbreviations are mentioned for the first time in the text. If the authors consider keeping the list of abbreviations, they should be arranged alphabetically and in a list or in a paragraph. They should initially mention the abbreviation and then define its meaning. For example: 

Abbreviations: AD, Alzheimer's disease; Aβ, Amyloid-β; ARS2, arsenite-resistance protein 2; ...

(6) Please review and correct Figures 1, 2, 3 and 4. Indicate all the abbreviations indicated in the figures, many of them are missing, e.g. in figure 1 there is no definition of interleukins. Please check and correct this problem in all figures.

- Figures 1, 2, 3 and 4: Abbreviations should be sorted alphabetically. Correct abbreviations, place only at the end and not at the first mention in the figure legends as they are indicated at the end. For example in Figure 4, they do not define what α-NETA and MIF mean, and Figure 4 should be in bold type. In Figure 4 the authors also mention "protein tyrosine phosphatases (PTPase)" and in the abbreviations they mention "PTPs: protein tyrosine phosphatases". Please correct this observation

- Figures 2, 3 and 4: Please correct these Figures. Place in the respective figure the numbers corresponding to the sequence of the pathway in the legends of each figure much larger to be able to better observe the pathway it indicates and to be able to follow the sequence better.

Inhibition, activation, promotion or interaction pathways should be added in the appropriate figure in the form of an image. In the legend of the figure, only what you have placed at the end should go "Meaning of symbols: →, promotion; -----|, inhibition; ↑, up-regulated; ↓,down-regulated; ~, interaction or binding". What was placed at the beginning as a sequence description should be removed.

(7) Page 2, line 58: The correct term is "NADPH oxidase", according to the authors "Orihuela, R.; McPherson, C.A.; Harry, G.J. Microglial M1/M2 polarization and metabolic states. Br. J. Pharmacol. 2016173, 649–665" as cited by Bianconi et al 2022.

(8) Table 1 should be corrected. I am enclosing a model in word format (look at compressed file) so that you can complete it with the data you have and be guided to make it clearer and more understandable. Of course this is a suggestion of the Table 1, you can take another option as well.

(9) Very important, authors should follow the Brain Sciences journal model for placing references, in this corrected version no changes are evident. For example:

- More than 10 authors, go with et al. (In the manuscript they place 6 and et al. which is not correct based on models of other papers published in Brain Sciences journal).

- The title of the paper should not be in italics.

- The year of publication is in bold and not the volume (which is in italics, but not in bold).

Please review and amend the list of references with the above changes.

Thank you very much for your time and cooperation

See compressed file for example and Table 1 in the ZIP file
